# Phylogenomics of white-eyes, a 'great speciator', reveals Indonesian archipelago as the center of lineage diversity

**Chyi Yin Gwee[1‡], Kritika M Garg[1†§], Balaji Chattopadhyay[1†#], Keren R Sadanandan[1,2], Dewi M Prawiradilaga[3], Martin Irestedt[4], Fumin Lei[5,6], Luke M Bloch[7], Jessica GH Lee[8], Mohammad Irham[3], Tri Haryoko[3], Malcolm CK Soh[9], Kelvin S-H Peh[10], Karen MC Rowe[11], Teuku Reza Ferasyi[12,13], Shaoyuan Wu[14,15], Guinevere OU Wogan[16], Rauri CK Bowie[7], Frank E Rheindt[1]***

[1]National University of Singapore, Department of Biological Sciences, Singapore, Singapore; [2]Max Planck Institute for Ornithology, Seewiesen, Germany; [3]Division of Zoology, Research Center for Biology, Indonesian Institute of Sciences (LIPI), Cibinong Science Center, Cibinong, Indonesia; [4]Department of Bioinformatics and Genetics, Swedish Museum of Natural History, Stockholm, Sweden; [5]Key Laboratory of Zoological Systematics and Evolution, Institute of Zoology, Chinese Academy of Sciences, Beijing, China; [6]Center for Excellence in Animal Evolution and Genetics, Chinese Academy of Sciences, Kunming, China; [7]Museum of Vertebrate Zoology and Department of Integrative Biology, University of California, Berkeley, Berkeley, United States; [8]Wildlife Reserves Singapore, Singapore, Singapore; [9]University of Western Australia, School of Biological Sciences, Perth, Australia; [10]University of Southampton, School of Biological Sciences, University, Southampton, United Kingdom; [11]Sciences Department, Museums Victoria, Melbourne, Australia; [12]Faculty of Veterinary Medicine, Universitas Syiah Kuala, Darussalam, Indonesia; [13]Jiangsu Key Laboratory of Phylogenomics and Comparative Genomics, School of Life Sciences, Jiangsu Normal University, Xuzhou, China; [14]Department of Biochemistry and Molecular Biology, 2011 Collaborative Innovation Center of Tianjin for Medical Epigenetics, Tianjin Key Laboratory of Medical Epigenetics, School of Basic Medical Sciences, Tianjin Medical University, Tianjin, China; [15]Center for Tropical Veterinary Studies – One Health Collaboration Center, Universitas Syiah Kuala, Darussalam, Indonesia; [16]Museum of Vertebrate Zoology and Department of Environmental Science, Policy, and Management, University of California, Berkeley, Berkeley, United States

**\*For correspondence:** dbsrfe@nus.edu.sg

[†]These authors contributed equally to this work

**Present address:** [‡]Division of Evolutionary Biology, Faculty of Biology, LMU Munich, Munich, Germany; [§]Institute of Bioinformatics and Applied Biotechnology, Electronics City, India; [#]Trivedi School of Biosciences, Ashoka University, Sonipat, India

**Competing interests:** The authors declare that no competing interests exist.

**Abstract** Archipelagoes serve as important 'natural laboratories' which facilitate the study of island radiations and contribute to the understanding of evolutionary processes. The white-eye genus *Zosterops* is a classical example of a 'great speciator', comprising c. 100 species from across the Old World, most of them insular. We achieved an extensive geographic DNA sampling of *Zosterops* by using historical specimens and recently collected samples. Using over 700 genome-wide loci in conjunction with coalescent species tree methods and gene flow detection approaches, we untangled the reticulated evolutionary history of *Zosterops*, which comprises three main clades centered in Indo-Africa, Asia, and Australasia, respectively. Genetic introgression between species permeates the *Zosterops* phylogeny, regardless of how distantly related species are. Crucially, we identified the Indonesian archipelago, and specifically Borneo, as the major center of diversity and

the only area where all three main clades overlap, attesting to the evolutionary importance of this region.

## Introduction

Archipelagoes are settings for unravelling complex evolutionary patterns as they constitute natural laboratories for the study of factors contributing to speciation, allowing for an examination of the evolution of lineages in isolation (*MacArthur and Wilson, 2001*; *Whittaker and Fernández-Palacios, 2007*; *Lohman et al., 2011*). Among vertebrate groups that occur across archipelagoes, island radiations of birds are most well-studied (*Lerner et al., 2011*; *Lamichhaney et al., 2015*). These avian models display a great deal of variability in their diversification rates across islands, which are fundamentally linked to species' capability to disperse over water (*Diamond et al., 1976*). In particular, the so-called 'great speciators', first characterized by *Diamond et al., 1976*, stand out from all other birds based on their paradoxical ability to disperse widely and colonize entire archipelagoes while, at the same time, diversifying into multiple daughter species in spite of a continuing potential for overwater gene flow (*Cai et al., 2020*).

One of the few classical examples of 'great speciators' identified by *Diamond et al., 1976* is the songbird genus *Zosterops*, or white-eyes, which are dispersers capable of differentiating rapidly from source populations (*Clegg et al., 2002*; *Moyle et al., 2009*). The genus *Zosterops* comprises c. 100 species that have radiated across the Old World and Oceania within the past 1–3.5 million years, reflecting one of the fastest diversification rates among vertebrates (*Warren et al., 2006*; *Moyle et al., 2009*; *Jetz et al., 2012*; *Leroy et al., 2019*; *Cai et al., 2020*). An overwhelming proportion (more than 70%) of these species occurs exclusively in archipelagoes distributed across the Atlantic, Indian, and Pacific Oceans (*Figure 1*). As such, the radiation of white-eyes serves as a model system with which to explore island biogeography theory (*Diamond et al., 1976*; *Moyle et al., 2009*).

In order to achieve an understanding of the underlying processes driving the white-eye radiation, its phylogeny first needs to be resolved to provide a reliable backbone for hypothesis testing. In *Zosterops*, however, traditional methods that rely on morphological tools to infer how species are related to one another have proven to be unreliable, as plumage features of ecologically distinct and geographically disjunct *Zosterops* species are often indistinguishable (*Mees, 1957*; *Mayr, 1965*). Although a more recent application of genetic methods has helped disentangle the white-eye radiation to some extent, most studies have concentrated on Afrotropical, Melanesian, and Indian Ocean members of the genus (*Slikas et al., 2000*; *Warren et al., 2006*; *Moyle et al., 2009*; *Cox et al., 2014*; *Linck et al., 2016*; *Wickramasinghe et al., 2017*; *Manthey et al., 2020*; *Martins et al., 2020*). There continues to be a dearth of knowledge on this radiation across the core of its Asian distribution due to limited sampling and lack of genetic data. In particular, it is crucial to unravel the phylogenetic affinities of white-eyes distributed across the Indonesian archipelago, which – comprising more than 17,000 islands – is the largest archipelago in the world and harbors about 20 endemic *Zosterops* species (*Figure 1*), including two species that were discovered in the last two decades and remain undescribed (*Eaton et al., 2016*; *O'Connell et al., 2019*). The high density of *Zosterops* species across the Indonesian archipelago hints at the possible importance of this region in white-eye evolution.

Apart from incomplete geographic sampling, the lack of resolution of the white-eye radiation has largely been a consequence of sparse genomic sampling: most phylogenetic studies of white-eyes have been restricted to one or a few genetic markers, resulting in trees that are plagued by unresolved polytomies, hampering useful evolutionary inference (*Slikas et al., 2000*; *Warren et al., 2006*; *Moyle et al., 2009*; *Oatley et al., 2012*; *Á.S and Joseph, 2013*; *Cox et al., 2014*; *Husemann et al., 2016*; *Linck et al., 2016*; *Round et al., 2017*; *Wickramasinghe et al., 2017*; *Shakya et al., 2018*; *Cai et al., 2019*; *Lim et al., 2019*; *O'Connell et al., 2019*; *Martins et al., 2020*). Disentangling relationships within rapid and recent radiations, such as white-eyes, requires overcoming the challenges of heterogeneous gene trees due to biological factors such as incomplete lineage sorting (*Edwards et al., 2005*; *Song et al., 2012*). The multispecies coalescent (MSC) model offers a promising avenue to overcoming gene tree discordance by allowing the evolutionary histories of each locus to be inferred independently (*Song et al., 2012*; *Liu et al., 2015*).

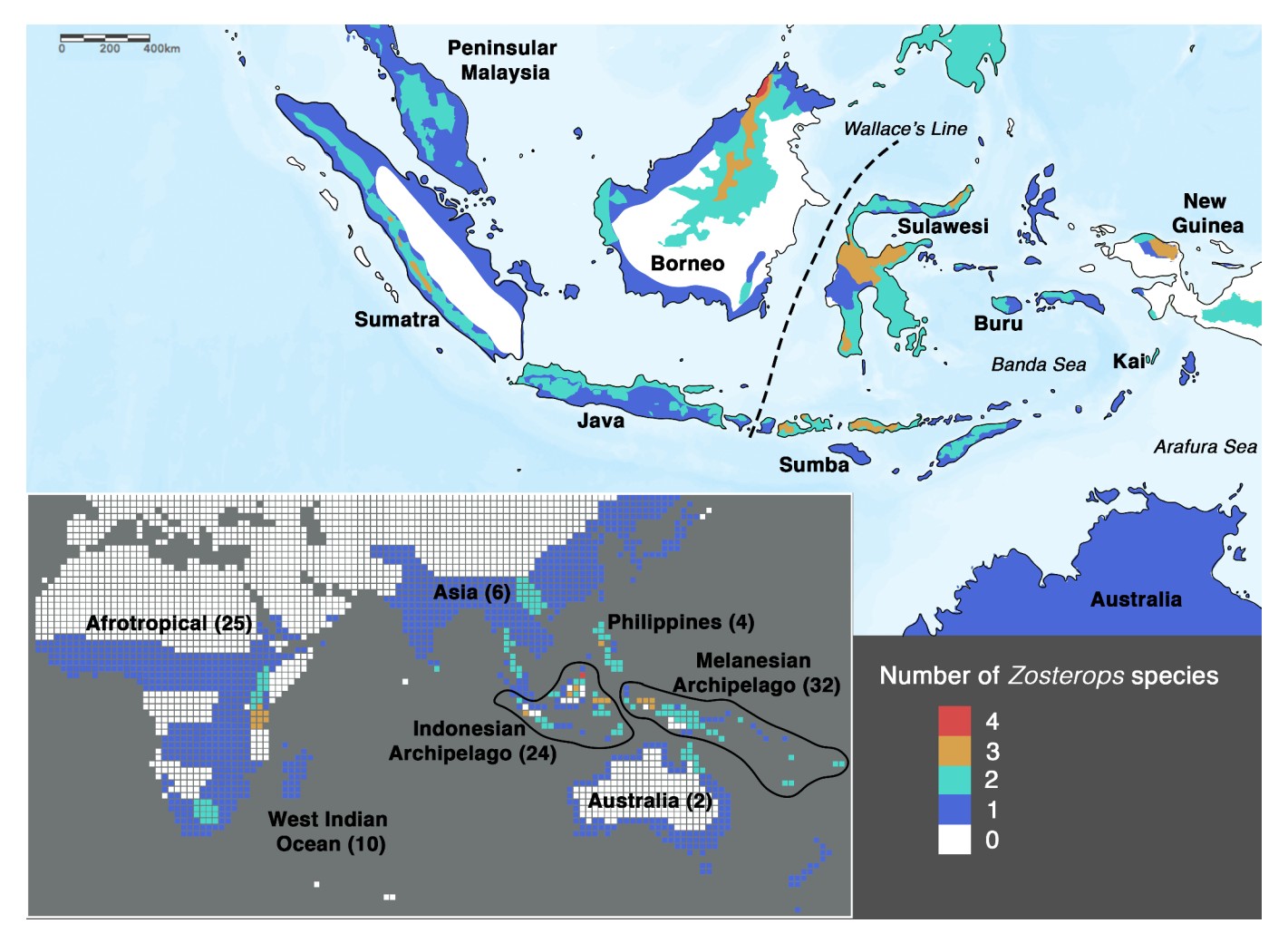

**Figure 1.** Species density map of the genus *Zosterops* across the Indonesian archipelago (main map) and across the entire distribution range (bottom left inset). Islands referred to in the text are specifically labeled on the main map. The total number of *Zosterops* species is shown in brackets beside each labeled region on the inset. We adopted *del Hoyo et al., 2016* as the baseline taxonomy and incorporated taxonomic revisions to the Afrotropical and Asiatic species as proposed by *Martins et al., 2020* and *Lim et al., 2019*, respectively (see *Supplementary file 2* for the list of recognized species).

An additional source of bias in reconstructing the phylogeny of rapid and recent radiations such as white-eyes is the potential for member species to engage in secondary gene flow, defined as post-speciation gene flow, or genetic introgression long after a speciation event has occurred (*Rheindt and Edwards, 2011*; *Edwards et al., 2016*). Such introgression will be reflected in the phylogenetic signal of a varying proportion of loci, thereby leading astray efforts to search for the true species tree. Multiple analytical approaches have been devised to account for secondary gene flow, such as through tree-based analysis as implemented in Phylogeographic Inference using Approximate Likelihoods (PHRAPL) (*Jackson et al., 2017a*; *Jackson et al., 2017b*), or through SNP-based analysis like the ABBA-BABA test, which detects an excess of shared derived alleles between populations (*Green et al., 2010*; *Patterson et al., 2012*).

In this study, we used historical specimens and recently collected samples to represent 33 white-eye species across the Southern hemisphere, especially from the understudied Indonesian archipelago (*Supplementary file 1*). We designed RNA probes using *Z. lateralis* (*Cornetti et al., 2015*) as a reference genome, targeting 832 loci at high coverage to overcome the limitations of missing data expected from degraded DNA of historical samples, thereby recovering a comparable set of loci

across both historical and fresh samples (*Templeton et al., 2013*). Our target capture methodology addresses the recalcitrant persistence of unresolved polytomies in the phylogeny of this rapid radiation by making use of a large set of loci (*Cai et al., 2019*). We employed three different coalescent species tree methods to assess topological incongruence across tree-building approaches (*Liu et al., 2009*; *Liu et al., 2010*; *Vachaspati and Warnow, 2015*). Recognizing that gene flow is commonly observed in recent radiations, we conducted PHRAPL (*Jackson et al., 2017b*) analysis and performed ABBA-BABA tests (*Patterson et al., 2012*) to assess introgression between closely related species with incongruent topologies and ultimately elucidate the likely evolutionary history of this complex radiation.

## Results

### Congruent phylogenetic trees reveal three distinct lineages

To shed light on the phylogenetic relationships of *Zosterops* species, we employed both concatenation methods, in which sequence data from individual loci are combined into one larger sequence, as well as MSC approaches (MP-EST [*Liu et al., 2010*], STAR [*Liu et al., 2009*], and ASTRID [*Vachaspati and Warnow, 2015*]), which account for individual gene tree stochasticity in a coalescent framework (*Edwards et al., 2007*). All four tree inference methods produced a similar phylogeny with a congruent tree topology for highly supported nodes, except for the placement of a Sundaic group consisting of *Z. atricapilla* and *Z. auriventer* (*Figure 2*). These two Sundaic taxa are embedded within the Australasian clade in the concatenated tree, but emerged within the Asiatic clade in the species trees constructed with MP-EST (henceforth our baseline species tree) and ASTRID (*Figure 2*). The other species tree method, STAR, shows a weak bootstrap support for an unresolved placement of these Sundaic taxa.

All methods reveal a white-eye radiation divided into three main clades consisting of an Indo-African, Asiatic, and Australasian group (*Figure 2*). Our taxon sampling covered 33 out of 108 white-eye species across the global radiation (*Supplementary file 2*). In order to expand clade assignment to well-studied white-eye species outside of our sampling regime, we examined an additional 30 species shown to have high bootstrap support for placement within any one of the three main clades based on previously published papers up until 2019 (*Warren et al., 2006*; *Moyle et al., 2009*; *Cox et al., 2014*; *Cornetti et al., 2015*; *Shakya et al., 2018*; *Cai et al., 2019*; *O'Connell et al., 2019*; *Figure 2—figure supplement 2*). These additional clade assignments were not used in the construction of our phylogenetic trees, but directly examined from the trees constructed by various studies. For example, *Cai et al., 2019* provide high bootstrap support (>90%) for the position of *Z. mouroniensis* as a descendant of the most recent common ancestor of two unequivocal members of the Indo-African clade (i.e., *Z. palpebrosus* and *Z. borbonicus*; *Figure 2*); thus the species is allocated accordingly, and the breeding distribution of *Z. mouroniensis* (Mt Karthala on Grande Comore Island) is shaded yellow on the global range map (*Figure 2a*).

A majority of the added taxa are distributed across Africa, where all *Zosterops* species fall within the Indo-African clade, while five of these newly added taxa are distributed across Melanesia, where our species coverage allowed us to detect the presence of only the Australasian clade. Therefore, our mapping suggests that the Afrotropical and the Australo-Papuan regions are depauperate in deeper *Zosterops* lineage diversity (*Figure 2a*; *Supplementary file 2*). Similarly, our results reveal that most areas in continental Asia generally harbor only one of the three main *Zosterops* clades, except East Asia where two clades co-occur in a narrow zone of overlap between the Indo-African *Z. palpebrosus* and the Asiatic *Z. simplex* (orange in *Figure 2a*). In contrast, the Indonesian archipelago emerged as a center of modern-day diversity for *Zosterops*, with all three main clades represented on Java and Borneo, and two main clades on many other islands (*Figure 2a*).

### Presence of secondary gene flow

The genetic signal of recent and rapid radiations is often convoluted by the presence of secondary gene flow, leading to heterogeneous gene trees which deviate from the true phylogeny. We assessed the presence of secondary gene flow specifically between members of a Sundaic species pair (*Z. auriventer* and *Z. atricapilla*) characterized by a shifting and incongruent placement across trees with representatives from the Asiatic (i.e. *Z. simplex*) and Australasian (i.e. *Z. emiliae* and *Z.*

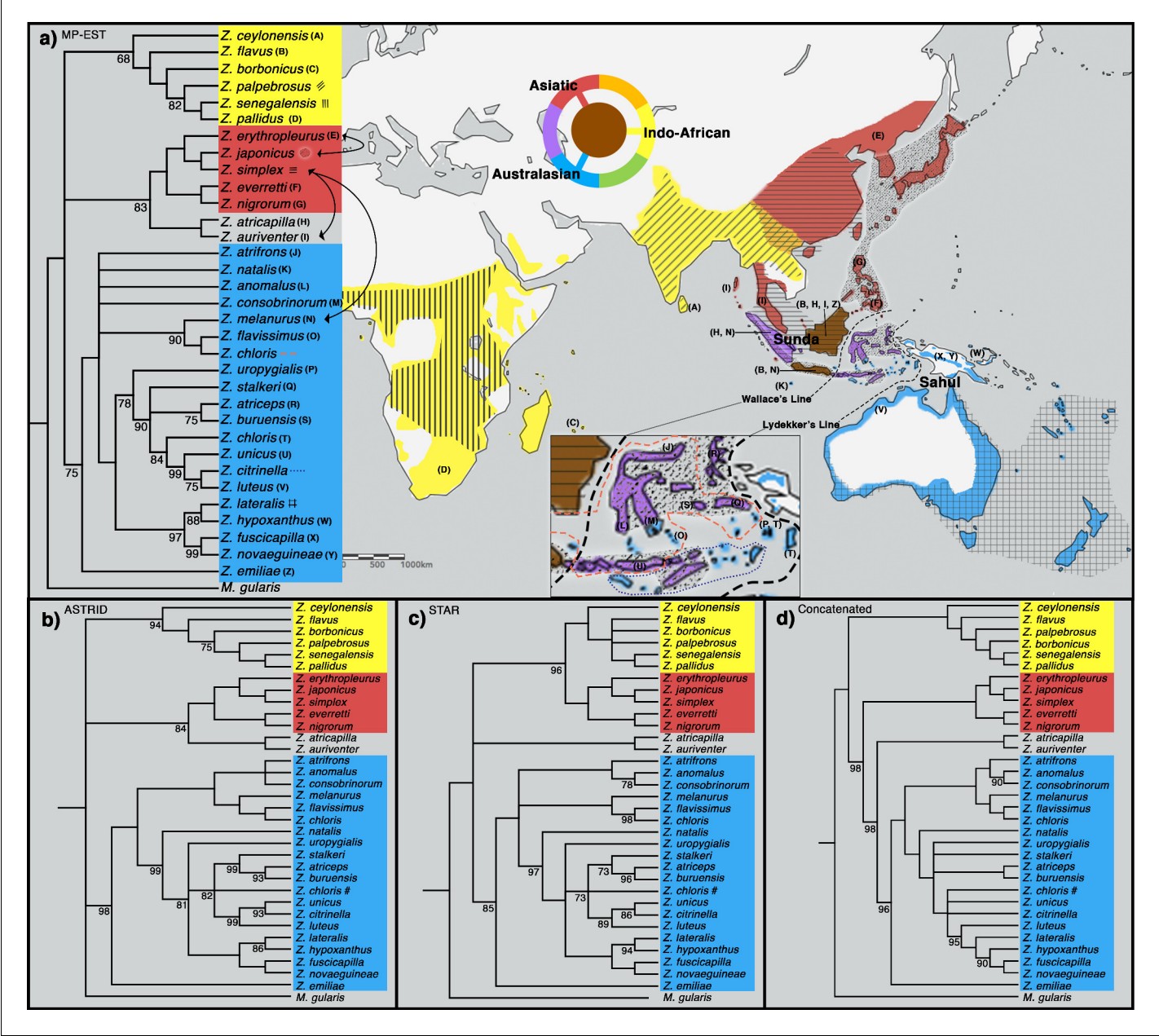

**Figure 2.** Phylogeny of *Zosterops*. Trees generated by (a) MP-EST, (b) ASTRID, (c) STAR and (d) the concatenation method (refer to *Figure 2—figure supplement 1* for a full concatenated tree and *Figure 2—figure supplement 2* for ancestral range estimation). All nodes are supported by a bootstrap value of 100 unless otherwise stated. Nodes with less than 68% bootstrap support were collapsed. The three main clades are color-coded blue (Australasian), red (Asiatic), and yellow (Indo-African). (a) The distribution of each main clade is color coded on the map, and the distribution of each sampled taxon is depicted by matching letter or symbol. The map includes 30 taxa not sampled by the present study but shown by previous studies to be nested within any of the three main clades with high bootstrap support of at least 90%. Borneo and Java (shaded brown) each harbor taxa from all three main clades, while multiple islands across the rest of Indonesia (shaded purple) each harbor taxa from two main clades. Secondary gene flow detected in multiple species pairs is marked with black arrows on the tree. Refer to *Figure 2—figure supplement 3* for a mitochondrial ND2 tree. The online version of this article includes the following figure supplement(s) for figure 2:

**Figure supplement 1.** Maximum likelihood tree run with RAxML based on 770 concatenated loci.

**Figure supplement 2.** Ancestral range estimation of the genus *Zosterops* using the DEC+j model in BioGeoBEARS on a concatenated maximum likelihood tree.

**Figure supplement 3.** Maximum likelihood phylogeny constructed with RaxML using a mitochondrial ND2 gene alignment of 1041 base pairs.

*melanurus*) clades (*Figure 3a*). The top two demographic models inferred by PHRAPL simulations show that Sundaic *Z. auriventer* is more closely related to Asiatic *Z. simplex* than to Australasian *Z. emiliae*, but inconclusive in relation to *Z. melanurus* due to ancestral and/or secondary gene flow between all three taxa (*Figure 3b*). Additionally, ancestral and/or secondary gene flow was detected between all taxa in both comparisons involving *Z. atricapilla*, *Z. simplex*, and either *Z. emiliae* or *Z. melanurus* (*Figure 3—figure supplement 1*). The genealogical divergence index (*gdi*) of the top demographic model inferred for each combination is relatively high, ranging between 0.321 and 0.751, suggesting high divergence between sister species despite secondary gene flow.

The ABBA-BABA approach corroborates that the conflicting placement of the Sundaic species pair consisting of *Z. atricapilla* and *Z. auriventer* may be attributed to secondary gene flow (*Figure 3*). In both concatenated and species trees, *Z. emiliae* emerges as basal to all other members of the Australasian clade (*Figures 2* and *3a*). Therefore, in the absence of introgression, the Sundaic *Z.*

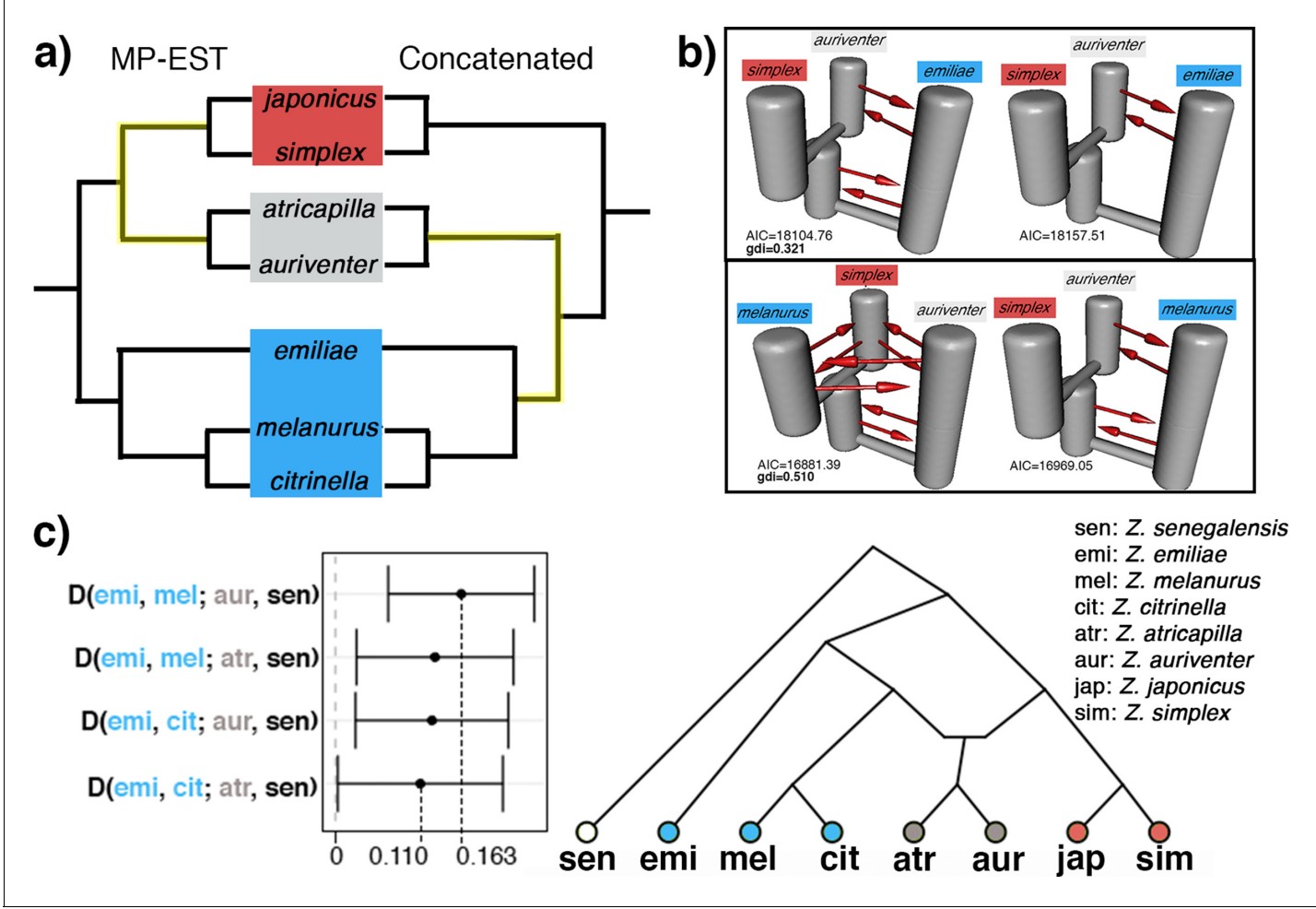

**Figure 3.** Detection of secondary gene flow in a Sundaic species pair of unresolved placement (gray) with members of either the Asiatic (red) or Australasian (blue) clade. (**a**) Placement of the Sundaic pair (*Z. atricapilla* and *Z. auriventer*) conflicts between MP-EST species tree and concatenated tree. (**b**) The top two demographic models in PHRAPL simulations show that Sundaic *Z. auriventer* is more closely related to Asiatic *Z. simplex* than to Australasian *Z. emiliae*, but inconclusive in relation to *Z. melanurus* due to secondary gene flow between the three taxa. Refer to *Figure 3—figure supplement 1* for simulation results with *Z. atricapilla*. (**c**) ABBA-BABA statistics for secondary gene flow shows an excess of derived allele sharing between the Sundaic taxa (gray) and Australasian *Z. melanurus* and *Z. citrinella*. (D-statistics significantly different from 0; see *Table 1* for full statistical results). Topology inferred from ABBA-BABA tests shows the two Sundaic lineages (*Z. atricapilla* and *Z. auriventer*) as carriers of genomic admixture between both Asiatic and Australasian clades.

The online version of this article includes the following figure supplement(s) for figure 3:

**Figure supplement 1.** Additional PHRAPL results.

*atricapilla* and *Z. auriventer* should be equally closely related to both *Z. emiliae* and other members of the Australasian clade (tree topology in *Figure 3a*). The ABBA-BABA statistics revealed a significant excess of allele sharing between the Sundaic taxa (*Z. atricapilla* and *Z. auriventer*) and two Australasian taxa (*Z. melanurus* and *Z. citrinella*) (*Figure 3c*, *Table 1*). This result reflects that ancestral introgression occurred between the Sundaic taxa and the Australasian lineage after *Z. emiliae* had diverged (*Figure 3c*).

In addition, secondary gene flow was also detected in several pairs of species which overlap in present-day distribution (sets 2 to 4 in *Table 1*). For instance, we found that *Z. melanurus* shares significantly more alleles with *Z. simplex* than with *Z. erythropleurus*, suggesting occasional hybridization between *Z. melanurus buxtoni* and *Z. simplex erwini* on Sumatra, where both occur. In a similar vein, *Z. erythropleurus* and *Z. japonicus* displayed excess allele sharing, indicating potential introgression in areas of overlap in Korea, and *Z. auriventer* and *Z. simplex* exhibited excess allele sharing that hints at occasional hybridization in parts of peninsular Malaysia and Borneo where they overlap.

## Discussion

### Phylogeny of *Zosterops* and presence of secondary gene flow

The evolutionary history of *Zosterops* has received a fair amount of scientific attention, but mostly by means of single mitochondrial or few nuclear loci, therefore resulting in trees plagued by unresolved polytomies (e.g. *Figure 2—figure supplement 3*; *Degnan and Moritz, 1992*; *Degnan, 1993*; *Slikas et al., 2000*; *Warren et al., 2006*; *Moyle et al., 2009*; *Oatley et al., 2012*; *Á.S and Joseph, 2013*; *Cox et al., 2014*; *Husemann et al., 2016*; *Linck et al., 2016*; *Round et al., 2017*; *Wickramasinghe et al., 2017*; *Shakya et al., 2018*; *Cai et al., 2019*; *Lim et al., 2019*; *O'Connell et al., 2019*). Using more than 700 genome-wide loci with a dense species sampling, our study produced an improved phylogeny of *Zosterops* and reveals the existence of three discrete main clades characterized by an Indo-African, Asiatic, and Australasian core of distribution, respectively (*Figure 2*). However, there was no strong support to unravel the sequence of diversification events among those three main clades. This lack of basal resolution could be attributed to a quick succession of divergence events at the time, and/or reticulated evolutionary history unresolved by species tree and concatenated tree approaches.

Both concatenated and species tree analyses resulted in congruent topologies for well-supported nodes, except for the placement of a Sundaic species pair comprising *Z. atricapilla* and *Z. auriventer* (*Figure 2*). Uncertainties in the affinity of these two Sundaic species appear to be due to secondary

**Table 1.** D-statistics of a selection of species combinations to test if two species (H2, H3) exhibit an excess of derived allele sharing. The ABBA-BABA test was restricted to species combinations with conflicting tree topologies observed in this study (set 1), conflicting tree topologies between this study and *Cai et al., 2019* (set 5), and a selection of species with a present-day geographic overlap and opportunities for secondary gene flow (sets 2–4). *Z. senegalensis* (H4) was used as an outgroup for all comparisons. A critical value (Z) above three suggests a significant excess of derived allele sharing between populations H2 and H3 and is highlighted in bold.

| Set | H1 | H2 | H3 | D-stat | Z | No. of ABBA | No. of BABA |
|---|---|---|---|---|---|---|---|
| 1 | *emiliae* | *melanurus* | *atricapilla* | 0.129 | **3.807** | 192.84 | 148.63 |
| | *emiliae* | *melanurus* | *auriventer* | 0.163 | **5.168** | 190.17 | 136.97 |
| | *emiliae* | *citrinella* | *atricapilla* | 0.110 | **3.086** | 178.55 | 143.09 |
| | *emiliae* | *citrinella* | *auriventer* | 0.125 | **3.788** | 172.21 | 133.99 |
| 2 | *citrinella* | *melanurus* | *simplex* | 0.093 | **3.074** | 142.55 | 118.22 |
| 3 | *simplex* | *japonicus* | *erythropleurus* | 0.215 | **5.272** | 171.87 | 111.15 |
| 4 | *erythropleurus* | *simplex* | *auriventer* | 0.108 | **3.169** | 166.20 | 133.74 |
| 5 | *melanurus* | *emiliae* | *simplex* | −0.053 | −1.456 | 152.33 | 169.34 |
| | *melanurus* | *emiliae* | *japonicus* | −0.066 | −1.729 | 149.05 | 170.02 |
| | *citrinella* | *emiliae* | *simplex* | 0.027 | 0.706 | 154.77 | 146.74 |
| | *citrinella* | *emiliae* | *japonicus* | −0.027 | −0.686 | 146.72 | 154.82 |

gene flow between the incipient Asiatic and Australasian clades (*Figure 3*). Such genetic introgression destabilizes the robustness of the MSC model, which does not account for secondary gene flow, thereby confounding species tree estimation (*Figure 3*, *Table 1*). At the same time, traditional phylogenetic approaches such as concatenation are equally negatively impacted by secondary gene flow and are additionally subject to the biases of incomplete lineage sorting (*Liu et al., 2019*).

To assess the magnitude of genetic introgression that has resulted in the controversial placement of the two Sundaic taxa, we used analytical approaches that specifically account for post-speciation gene flow, such as PHRAPL. Most of the top demographic models inferred by PHRAPL produced topologies in which the Sundaic species of controversial placement emerge as sister to the Asiatic clade (*Figure 3b*, *Figure 3—figure supplement 1*). However, conflicting demographic models were observed between the top two results of some combinations due to ancestral and/or secondary gene flow in all tested directions (*Figure 3*, *Figure 3—figure supplement 1*). For instance, the erroneous inference of *Z. melanurus* and *Z. simplex* emerging as sister species to each other – basal to *Z. auriventer* – may be attributed to secondary gene flow also present between *Z. melanurus* and *Z. simplex* (*Figure 3b*, *Table 1*). Convergence onto a single demographic model may require further PHRAPL simulations allowing for more parameters such as asymmetrical rates of gene flow (*Morales et al., 2017*). Additional demographic analyses using other programs, such as DaDi (*Gutenkunst et al., 2010*) and fastsimcoal (*Excoffier and Foll, 2011*), may also assist in fully disentangling the complex relationships within this rapid radiation.

SNP-based analysis using ABBA-BABA statistics conclusively identified introgression as an underlying cause of the conflicting placement of *Z. atricapilla* and *Z. auriventer*. Specifically, excess allele sharing between these two species and *Z. melanurus*, but not between them and *Z. emiliae*, suggests introgression between the incipient stages of the Asiatic and Australasian clades – after *Z. emiliae* had split off from other Australasian species (*Figure 3c*). Such ancient introgression generates patterns of allele sharing that would lead to the two controversial Sundaic species partly being reflected as members of the one clade or of the other (*Figures 2* and *3*), depending on tree inference methods.

More generally, the ABBA-BABA test detected rampant secondary gene flow between species that geographically overlap, regardless of their phylogenetic proximity, underscoring the pervasive nature of genetic introgression in rapidly evolving lineages such as white-eyes (black arrows in *Figure 2a*; sets 2–4 in *Table 1*). For example, *Z. auriventer* and *Z. simplex* display excess allele sharing indicating recent gene flow in areas of Sundaland where they overlap, even though they belong to different main clades of *Zosterops* (*Figure 2*, *Table 1*). The same is true for *Z. melanurus* and *Z. simplex*, which co-occur on Sumatra, and for *Z. japonicus* and *Z. erythropleurus*, which overlap in Korea (*Table 1*). Such introgression between species is likely recent, limited, and of the kind that the MSC model remains robust to *Liu et al., 2009*. The detection of rampant secondary gene flow in multiple pairs of sympatric white-eye species is in agreement with the recent discovery of introgression between various non-sister white-eye species across the Solomon Islands (*Manthey et al., 2020*).

## Indonesian archipelago harbors all three main clades

While the highest rates of *Zosterops* diversification are known to have occurred on archipelagoes in general (*Diamond et al., 1976*; *Moyle et al., 2009*), the geographic distribution of deeper-level lineage diversity in this genus remains unexplored. Our phylogenetic results demonstrate that Africa, most parts of continental Asia and probably also all of the Australo-Papuan region each harbor representatives from only one of the three main *Zosterops* clades, respectively (*Figure 2a*), regardless of *Zosterops* species diversity. In contrast, virtually all parts of the Indonesian archipelago harbor white-eye species from two to three of the main *Zosterops* clades (*Figure 2a*). Positioned between the Sunda and Sahul shelf, the Indonesian islands are a center of syntopy of lineages from either side of Wallace's Line (*Moss and Wilson, 1998*; *de Bruyn et al., 2014*). This is consistent with the rapid rate of tectonic change reconstructed for the Indonesian archipelago over the last 30 million years (*Hall, 2002*; *Hall, 2012*; *de Bruyn et al., 2014*; *Nugraha and Hall, 2018*), which has led to a narrowing of open sea between Asia and Australia and thereby facilitated overwater dispersal of Sundaic and/or Australo-Papuan lineages for many organismic groups (e.g. *Heads, 2001*; *Irestedt et al., 2013*; *Gwee et al., 2017*; *Ng et al., 2017*; *Garg et al., 2018*; *Reilly et al., 2019*; *Oliver et al., 2020*, including white-eyes). At the same time, while the narrowing of the sea gap

between Australo-Papua and Asia has created numerous stepping-stone islands to facilitate overwater dispersal, most of the Wallacean region remains dominated by deep sea, and there are as yet no land bridges (*Voris, 2000*; *Rheindt et al., 2020*). This complicated archipelagic setting has likely acted as a diversification driver in white-eyes, which have the capability of colonizing and populating these deep-sea islands.

To the west of Wallace's Line, the Sundaic islands of Borneo and Java, which constitute a large part of the Greater Sunda archipelago, each harbor a number of representatives of the Asiatic and Australasian clades but are additionally inhabited by the coastal endemic species *Z. flavus*. This species is phylogenetically more closely related with the Indo-African clade rather than with the geographically more proximate Asiatic and Australasian clades, demonstrating an impressive potential for dispersal capability that may in part underlie the rapid diversification rate of the genus.

Low-lying Indonesian islands to the east of Wallace's Line, such as Sumba and Kai, generally harbor only *Zosterops* species from the Australasian clade, whereas *Z. japonicus* of the Asiatic clade additionally occurs on islands that reach montane elevations of over 1200 m. For example, *Z. japonicus* is present on mountainous Buru (c. 9500 km$^2$ in size) but absent on Sumba (c. 11,000 km$^2$ in size), which largely comprises savannah with a small hilly region not exceeding 1200 m in elevation, despite Sumba being a larger island (*Figure 1*). Although the elevation of an island contributes substantially to the number of main *Zosterops* clades present, it seems to have less influence on the total number of *Zosterops* species (*Figure 1*). For instance, the Kai islands, with a combined area of only approximately 1400 km$^2$ and an elevation of less than 700 m, harbor three distinct *Zosterops* species, including two island endemics *Z. uropygialis* and *Z. grayi* (not sampled) (*Figures 1* and *2*).

Our phylogenetic results reveal that the widespread Lemon-bellied White-eye *Z. chloris* is non-monophyletic (*Figure 2*; *Figure 2—figure supplement 1*) and several small, low-lying islands situated between the Banda Sea and Arafura Sea, including Kai and Aru, may harbor a cryptic species morphologically identical to the Lemon-bellied White-eye. White-eyes are renowned for their conservative morphology, which contributes to the confusion that has surrounded their taxonomic treatment (*Mees, 1957*; *Mayr, 1965*; *Lim et al., 2019*; *Manthey et al., 2020*). Further research is required to ascertain the evolutionary status of these and other overlooked island populations.

## Borneo is a hotspot for evolution and harbors deep phylogenetic isolates

Borneo has been identified as a major source of diversification across the Southeast Asian region for a variety of organismic groups, including birds, mammals, amphibians, and plants (*de Bruyn et al., 2014*). Our study reveals Borneo's unique status as the only place in the distribution of the genus where members of all three main *Zosterops* clades occur, and where as many as four *Zosterops* species co-exist within a few square kilometers of one another, rendering it the center of faunal mixing for white-eyes (*Figures 1* and *2*). Borneo forms the eastern part of the Sundaic region, which – at present – is splintered into multiple bigger and many smaller landmasses comprising the Greater Sunda Islands and Malay Peninsula. For the longest time over the past 400,000 years, however, these landmasses have been merged into a larger landmass, Sundaland, that has facilitated the evolution of much of Southeast Asia's equatorial rainforest fauna (*Sarr et al., 2019*). The east of Sundaland (i.e. Borneo) has constituted a particularly stable part of this subcontinent, remaining above water for the longest uninterrupted time and offering a wide variety of habitats such as mangroves (*Z. flavus*), submontane forest (*Z. atricapilla* and *Z. auriventer*), montane forest (*Z. emiliae* and a still undescribed white-eye from the Meratus range *Eaton et al., 2016*), and coastal woodland (*Z. simplex*). Our results from ancestral range estimation suggest an Asian origin for the entire *Zosterops* radiation, and a Sundaic origin for the Australasian clade (*Figure 2—figure supplement 2*). Borneo constitutes the largest landmass within the Sundaic region, and its elevated count in discrete *Zosterops* lineages suggests that it has played a key role in the diversification of this important radiation.

While Borneo's exceptional biodiversity has been appreciated as early as during Alfred R. Wallace's times (*Wallace, 1962*), most of this diversity has traditionally been interpreted as being of a Sundaic element largely shared with Sumatra and peninsular Malaysia, and is only slowly being recognized as having attained species-level depths of differentiation (*Cros et al., 2020*). On the other hand, Borneo is known for hosting a number of deep phylogenetic isolates, such as the enigmatic Bristlehead *Pityriasis gymnocephala* (*Oliveros et al., 2019*). In the context of *Zosterops* diversification, we add the Mountain Black-eye *Z. emiliae* as an overlooked phylogenetic isolate (*Figure 2*).

Although it has been shown to be embedded within *Zosterops* for over a decade (*Moyle et al., 2009*; *Cai et al., 2019*), it continues to be treated as a monospecific genus (*Chlorocharis*) by some modern sources (*del Hoyo et al., 2016*). Using our much-improved taxon sampling, *Z. emiliae* emerged as a basal sister to the Australasian clade with moderate support under various analytical regimes (*Figure 2*). In contrast, *Cai et al., 2019* placed *Z. emiliae* with members of the Asiatic clade. Our use of the ABBA-BABA test to verify whether this conflicting position may be due to ancient introgression did not identify an excess of allele sharing between *Z. emiliae* and the Asiatic members (set five in *Table 1*), suggesting that the incongruent placement of *Z. emiliae* by *Cai et al., 2019* is unlikely to be a result of secondary gene flow. Instead, incomplete lineage sorting may have generated such phylogenetic conflict as the divergence of *Z. emiliae* likely fell within a time of rapid diversification within the genus, leading to its recalcitrance to phylogenetic resolution when only few loci are applied.

## Conclusions

Our study presents the application of species tree methods on a large set of genome-wide markers across a comprehensive sampling of members of a rapid radiation of a classic 'great speciator'. Using approaches to account for secondary gene flow, our study demonstrates the pervasive presence of genetic introgression across this explosive radiation. The resultant phylogeny of *Zosterops* white-eyes reveals that the Indonesian archipelago, and Borneo in particular, are an evolutionary hotspot for the diversification of the genus. This archipelagic region harbors members of clades centred in the neighboring Asian and Australo-Papuan landmasses, and even from a geographically distant Indo-African clade. The western Indonesian archipelago is the sunken remnant of a subcontinent – Sundaland – that only started to be periodically submerged starting from ~400,000 years ago, and offers potential for differentiation. The identification of areas in western Indonesia as a major center of modern phylogenetic diversity not only contributes to their recognition as an arena of important evolutionary processes, but also elevates their status as a region of global conservation relevance.

## Materials and methods

### Taxon sampling

A total of 48 historical toepad samples and 52 fresh samples were acquired from various museums and through fieldwork conducted across peninsular Malaysia and the Indonesian archipelago (*Ashari et al., 2019*; *Supplementary file 1*). In total, 33 white-eye species were represented [following the taxonomy by *del Hoyo et al., 2016* with more recent revisions by *Lim et al., 2019* and *Martins et al., 2020* (see *Supplementary file 2*)].

### Probe design for target capture

Target enrichment protocols have been shown to be highly effective at capturing historical DNA for phylogenomic studies (*Bryson et al., 2016*; *van der Valk et al., 2017*; *Chattopadhyay et al., 2019*; *Baveja et al., 2020*). We designed loci specifically targeting both conserved exons and variable intronic regions of the *Zosterops* genome (*Chattopadhyay et al., 2019*). We first used EvolMarkers (*Li et al., 2012*) to identify conserved single copy coding sequences in the genomes of *Z. lateralis* (accession no. GCA_001281735) (*Cornetti et al., 2015*), *Ficedula albicollis* (accession no. GCA_000247815.1) (*Ellegren et al., 2012*), and *Taeniopygia guttata* (accession no. GCF_003957565.1; released by the Vertebrate Genomes Project). To identify conserved exons, EvolMarkers performs a BLAST search (*Altschul et al., 1990*), for which we set a minimum of 55% identity and e-value of less than 10E-15. Only single-hit exons longer than 500 bp were used for further downstream analysis. Then we isolated 500 bp upstream and downstream of these conserved exons from the *Z. lateralis* genome to include variable intronic regions using bedtools 2.28.0 (*Quinlan and Hall, 2010*). We further checked for overlapping targets and merged all overlapping loci in bedtools, removing any loci with GC content less than 40% or more than 60%. Loci comprising repeat elements were identified using RepeatMasker 4.0.7 (*Smit et al., 2015*) and removed. After filtering, our design retained 832 loci, which were used by Arbor Biosciences (USA) to design a total of 63,244 RNA baits. Each locus was targeted with 4X tiling density of overlapping baits, each bait of 100 bp, for in-solution target enrichment.

## Laboratory procedures

Fresh DNA was extracted following the manufacturer's protocol using the DNeasy Blood and Tissue Kit (Qiagen, Germany). The DNA of historical toepad samples was extracted under sterile conditions inside a dedicated ancient DNA facility, and extractions were performed inside a biosafety cabinet with laminar air-flow. The ancient DNA facility room was subject to at least 12 hr of UV light and thoroughly cleaned with bleach in between each session of historical DNA extractions. We used the same kit for extraction of historical DNA with slight modifications (*Chattopadhyay et al., 2019*). Extraction negatives were included to ensure absence of contamination. Double-stranded DNA concentrations were ascertained using a Qubit 2.0 high sensitivity DNA Assay kit (Invitrogen, USA), and fragment sizes were assessed using an AATI Fragment Analyzer (Agilent, USA). The negatives were also quantified using a Qubit 2.0 assay and AATI to ensure absence of DNA.

Fresh DNA was sheared into a targeted size of 250 bp using a Bioruptor Pico sonication device (Diagenode, Belgium) with 13 cycles of sonication prior to library preparation. Each cycle consisted of 30 s of sonication followed by 30 s of rest. We used NEBNext Ultra II DNA Library Prep Kits for Illumina (New England BioLabs, USA) and NEBNext 8 bp dual indexes (New England BioLabs, USA) for both fresh and historical library preparation. The libraries using fresh tissue were size selected for an insert size of 250 bp with AMPure XP (Beckman Coulter, USA) beads, giving an expected final library size of ~370 bp with adapters and primers included. Size selection was omitted during library preparation of historical samples to reduce DNA loss. A total of 12 cycles of polymerase chain reaction (PCR) was applied and negative controls were carried out for both fresh and historical libraries.

The library preparation of historical DNA was conducted inside a dedicated PCR cabinet with laminar air-flow. The PCR cabinet was subject to 1 hr of UV light and thoroughly cleaned with bleach in between each batch of library preparations. We added NEBNext FFPE DNA repair mix (New England BioLabs, USA) to the historical DNA prior to library preparation to reduce deamination of cytosine to uracil, repair nicks, and fill in 5' overhangs of the damaged DNA. DNA quantification and assessments of libraries' fragment sizes were conducted as above. The peak fragment sizes of each library prepared with historical samples ranged between 200 bp and 300 bp, whereas the peak fragment sizes of each library prepared with fresh samples ranged between 330 bp and 420 bp. The negatives were also quantified by Qubit 2.0 and assessed by AATI to ensure that only adapters and primer-dimer DNA (single peak at ~55 bp and ~150 bp respectively) were present.

Target capture was performed on all historical and fresh samples using a MYbaits kit version 3 (Arbor Biosciences, USA), with a modified protocol following *Chattopadhyay et al., 2019*. In brief, we diluted the volume of baits to 1.85 µL per historical DNA sample (~3X dilution) and 1.1 µL per fresh DNA sample (~5X dilution). Biotinylated RNA baits and target sequences were hybridized at 60°C for 40 hr for historical samples and at 65°C for 20 hr for fresh samples. Following hybridization, the samples were cleaned according to the myBaits manual, and PCR was conducted using IS5 and IS6 primers with 20 cycles for historical samples and 15 cycles for fresh samples (*Fortes and Paijmans, 2015*). DNA quantification and assessments of libraries' fragment sizes were conducted as above (see extraction). The peak fragment sizes of each historical target capture library ranged between 250 bp and 400 bp, whereas the peak fragment sizes of each fresh target capture library ranged between 370 bp and 410 bp. The target capture libraries were sequenced using the Illumina HiSeq 2500 and HiSeq 4000 platforms with 150 bp paired-end runs for all samples. Fresh and historical samples were run on separate lanes.

We additionally sequenced the whole genomes of nine *Zosterops* individuals. Their DNA was extracted using the DNeasy Blood and Tissue Kit (Qiagen, Germany) according to the manufacturer's protocol. The samples were then prepared using a Nextera Library Prep Kit (Illumina, USA) with dual indexes. The whole genome libraries were sequenced on the Illumina X10 or NovaSeq platforms at Medgenome (Foster City, California) with 150 bp paired-end runs.

## Sequence assembly

We removed adapter sequences using Trimmomatic 0.38 (*Bolger et al., 2014*) and duplicates with FastUniq 1.1 (*Xu et al., 2012*). Paired trimmed reads of historical samples were examined with map-Damage 2.0.9 (*Jónsson et al., 2013*) to assess DNA deamination in read ends. We further trimmed 5 bp from the 3' ends of both forward and reverse reads as the mapDamage results show a high amount (>0.1 probability) of G to A misincorporation in the read ends, and reran mapDamage to

ensure the average probability of misincorporation across samples remained below 0.1. We ran HybPiper 1.3.1 (*Johnson et al., 2016*) to extract target sequences. Following the HybPiper pipeline, the trimmed reads were aligned to each target gene using BWA 0.7.17 (*Li, 2013*). We conducted a de novo assembly of target sequences using SPAdes 3.13 (*Bankevich et al., 2012*), applying a sequencing depth cut-off of at least 16X coverage per contig. The contigs generated by SPAdes were re-aligned against the target sequences using Exonerate 2.4.0 (*Slater and Birney, 2005*) to assemble coding sequence regions (including intronic regions) and the resulting DNA sequence of each locus was extracted for downstream analyses. The length of each locus assembled for each sample was examined using the following python scripts in the HybPiper package: get_seq_lengths.py and hybpiper_stats.py. We removed four historical samples due to high amount of missing data and kept 770 loci after visual quality checks across all samples, ensuring each locus is present in at least 85% of individuals and contains less than 30% missing nucleotides.

An additional 12 *Zosterops* samples were included by extracting target sequences from their whole genomes using blastn in BLAST+ 2.6.0 (*Camacho et al., 2009*). The whole genomes of *Z. lateralis* (accession no. GCA_001281735.1) (*Cornetti et al., 2015*), *Z. pallidus* (accession no. GCA_007556475.1) (*Leroy et al., 2019*), and *Z. borbonicus* (accession no. GCA_007252995.1) (*Leroy et al., 2019*) were obtained from NCBI. Whole genomes of nine individuals were resequenced for this study (*Supplementary file 1*). Target sequences were also extracted from the *Mixornis gularis* whole genome (*Tan et al., 2018*) to be used as an outgroup for some downstream analyses. For the nine whole genomes that were generated for the present study, we first cleaned the raw reads using a modified Perl script to remove exact PCR duplicates and low complexity reads (*Bi et al., 2012*; *Singhal, 2013*), and used Trimmomatic 0.30 (*Bolger et al., 2014*) for adapter removal. Contaminants were removed by aligning the raw reads to the reference genomes of potential contaminant sources (such as ribosomal RNA, human, and bacterial DNA) with Bowtie 2.0.1 (*Langmead and Salzberg, 2012*), followed by another cleanup of reads using Cutadapt 1.16 (*Martin, 2011*). We merged overlapping paired reads using FLASH 1.2.11 (*Magoč and Salzberg, 2011*), and aligned the cleaned paired reads against the *Zosterops lateralis* genome using BWA-MEM in BWA 0.7.8 (*Li, 2013*). The data were then converted into bam file format and sorted using SAMtools 1.5 (*Li et al., 2009*). We used mpileup in BCFtools (*Li et al., 2009*) to calculate the genotype likelihoods of each site of the reads, and generated a consensus sequence in fasta format using BCFtools. For all 12 whole genome resequenced samples, we used blastn in BLAST+ 2.6.0 (*Camacho et al., 2009*) to extract the same set of loci as that used in target capture.

## Phylogenetic analyses

MAFFT 7.0 (*Katoh and Standley, 2013*) was run to ensure each locus direction was consistent throughout all samples. All 770 loci were concatenated, giving a final alignment of 1,635,155 bp with 7.60% gaps. RAxML 8.2.12 (*Stamatakis, 2014*) was run to construct a Maximum Likelihood (ML) tree using a GTR + GAMMA + Invariant Sites model with 100 rapid bootstraps to search for the best-scoring ML tree, and the tree was rooted with *M. gularis*.

Best-scoring ML gene trees were inferred for each locus with RAxML 8.2.12 (*Stamatakis, 2014*) with 20 independent searches from a random starting tree and using a GTR + GAMMA + Invariant Sites substitution model. Each gene tree was run with 100 bootstraps for node support. A total of 768 gene trees were rooted with *M. gularis*, while no outgroup sequence was present for two gene trees which had to be discarded for species tree inference.

We adopted the MSC model using three different algorithm methods: MP-EST 2.0 (*Liu et al., 2010*), STAR (*Liu et al., 2009*), and ASTRID 1.4 (*Vachaspati and Warnow, 2015*). All three species tree methods were run with the 768 best-scoring ML gene trees to infer the species tree topology, and with 100 different sets of input gene trees to infer bootstrap support. Nodes with less than 68% bootstrap support were collapsed.

We assessed the level of congruence in the phylogenetic placement of *Zosterops* species between the present and previously published datasets following the species-level classification by *del Hoyo et al., 2016* and the more recent taxonomic revisions within the Asiatic and Afrotropical white-eye complexes by *Lim et al., 2019* and *Martins et al., 2020*, respectively (see *Supplementary file 2*). Specifically, we assigned species to one of three main clades that emerged within the genus in our analyses. We expanded our clade assignment to 30 *Zosterops* taxa not sampled in our study but found to be embedded within one of the three main clades with high

bootstrap support (>90%) by at least one study up until 2019 (*Supplementary file 2*). These additional clade assignments were not used in the construction of our phylogenetic trees, but directly examined from the trees constructed by the respective studies indicated in *Supplementary file 2*.

## Mitochondrial tree

We observed non-specific binding during hybridization of probes and sample DNA, allowing us to assemble mitochondrial DNA from the raw reads of each individual prepared by target capture. The raw reads were first mapped to the reference mitogenome of *Z. lateralis* (accession no. NC029146) using bwa 0.7.17 (*Li, 2013*), and converted to bam files using SAMtools 1.9 (*Li et al., 2009*). The bam files were then imported into CLC Genomics Workbench 7.0.4, remapped to the same reference mitogenome, and locally re-aligned. A consensus mitogenome of each individual was extracted with a minimum coverage of five, otherwise an ambiguous base 'N' was inserted. Finally, we extracted 1041 bp of ND2 sequence from each individual by aligning each assembled mitogenome to the ND2 sequence of *Z. lateralis*. Some samples were removed due to extensive missing nucleotides, and the ND2 sequences of 68 individuals were retained. As a means to assess the presence of artifacts from DNA damage, especially in toepad DNA, we compared the ND2 sequences generated in this study with the ND2 sequences of the same taxa deposited on GenBank by previously published studies. We also added the Genbank sequences of 16 *Zosterops* species not represented in our sampling regime. A maximum likelihood tree was generated using RAxML with 10,000 bootstrap replicates under the GTR + GAMMA model.

## Testing gene flow with ABBA-BABA statistics

We computed SNP-based ABBA-BABA statistics to test for gene flow among a subset of taxa with topological incongruence among trees, as well as populations which overlap in distribution. The bam files generated during locus assembly were used as input for SNP calling for the fresh samples, while the bam files of the historical samples were first subjected to mapDamage 2.0.9 (*Jónsson et al., 2013*) to rescale the quality scores of possibly deaminated sites. We used ANGSD 0.923 (*Korneliussen et al., 2014*) to call SNPs with the following filters applied: minimum depth of 20, block size of 50,000 bp, remove transitions, minimum mapping quality of 30, and minimum base quality of 20. We computed D statistics and used the jackknife.R script in ANGSD 0.923 (*Korneliussen et al., 2014*) to compute critical values (Z) and test for significance. A positive critical value $Z > 3$, corresponding to a p-value below 0.0013, suggests a significant excess of ABBA-like alleles as compared to BABA-like alleles (*Green et al., 2010*; *Patterson et al., 2012*). This critical value is widely applied as a threshold for detecting introgression and reduces the likelihood of false positives (*Zheng and Janke, 2018*).

## Demographic analysis

We additionally assessed the presence of secondary gene flow using PHRAPL 0.62 (*Jackson et al., 2017b*) by simulating the probability of observing a set of gene trees across various demographic models. We conducted simulations on a smaller subset of populations exhibiting topological incongruence: *Z. auriventer* (A), *Z. atricapilla* (B), *Z. simplex* (C), *Z. emiliae* (D) and *Z. melanurus* (E). A total of four combinations (ACD, ACE, BCD and BCE) were tested with 770 gene trees, each dataset consisting of three populations and an outgroup, *Z. senegalensis*. Each population was randomly subsampled to at most four individuals with ten replicates per locus: four out of five *Z. auriventer* individuals, four out of ten *Z. simplex* individuals, four out of seven *Z. melanurus* individuals, one out of two *Z. atricapilla* samples, and one *Z. emiliae* individual. We generated 48 possible demographic models under the following settings: an overall maximum of three free parameters (K = 3), a maximum of two coalescent events (K = 2), either complete isolation or migration event(s) with a single rate (K = 1), no variation in population size and growth (K = 1), only fully resolved trees were assumed, and only symmetrical migration between populations was assumed. Each dataset was simulated under these 48 different models with 10,000 trees using grid search. The probability of observing the set of gene trees under each model was assessed using Akaike Information Criterion (AIC) scores and the top two models with the lowest AIC values for each dataset were plotted in R 3.6.1 (*R Development Core Team, 2019*). We included the second-best model for each combination in our assessment because a previous study has shown that the second-best model may emerge

as the correct demographic model with further testing (*Morales et al., 2017*). Additionally, we computed the genealogical divergence index (*gdi*) of each combination to assess the overall level of divergence between the sister taxa inferred from each model with the combined effects of genetic drift and gene flow (*Jackson et al., 2017a*). A *gdi* index close to 0 suggests panmixia, while an index close to one suggests strong divergence.

### Ancestral range estimation

We ran BioGeoBEARS 1.1.2 (*Matzke, 2014*) in R 3.6.1 (*R Development Core Team, 2019*) to estimate the ancestral range of each internal node of the unrooted ML concatenated tree. We used ETE Toolkit 3.1.1 (*Huerta-Cepas et al., 2016*) python script to set the root node between the Indo-African clade and the other clades. We assigned each species to one or more out of ten geographical areas: the Afrotropical region, West Indian Ocean islands, South Asia, mainland Southeast Asia, East Asia, Philippines, the Sundaic region, Wallacea, Melanesia, and Australia. We applied all six models (DEC, DEC+j, DIVA, DIVA+j, BayArea and BayArea+j) and selected the best-fitting model (DEC+j) based on AIC values.

## Acknowledgements

We are indebted to the following staff and museums for loaning toepads and tissue material to us: P Sweet (American Museum of Natural History, New York, USA); A Drew and L Joseph (Australian National Wildlife Collection – Commonwealth Scientific and Industrial Research Organisation, Canberra, Australia); SM Birks (Burke Museum of Natural History and Culture, Seattle, USA); KP Lim (Lee Kong Chian Natural History Museum, Singapore); Museum of Vertebrate Zoology (Berkeley, USA); Museums Victoria (Melbourne, Australia); Museum Zoologicum Bogoriense (Bogor, West Java, Indonesia); P Kamminga (Naturalis Biodiversity Center, Leiden, Netherlands); M Penck (South Australian Museum, Adelaide, Australia); J Stigenberg and U Johansson (Swedish Museum of Natural History, Stockholm, Sweden); Western Australian Museum (Perth, Australia); and K Zyskowski (Yale Peabody Museum of Natural History, New Haven, USA). The Ministry of Research and Technology of Indonesia issued the following research permits: 10/TKPIPA/FRP/SM/X/2013 to FER; 314/SIP/FRP/E5/Dit. KI/X/2018 to FER and CYG; and 233/SIP/FRP/SM/VI/2013 and 147/SIP/FRP/SM/V/2014 to JM McGuire and RCKB. The Economic Planning Unit of Malaysia issued the research permit UPE 40/200/19/3295 to MCKS and KS-HP. We thank BTM Lim, EYX Ng, P Baveja, YF Chung, RYC Teo, E Arida, KC Rowe and JM McGuire for field, lab and/or analytical support, and the reviewers for their valuable comments. KMG acknowledges support from the Ramanujan Fellowship of SERB.

## Additional information

### Funding

| Funder | Grant reference number | Author |
|---|---|---|
| Ministry of Education - Singapore | R-154-000-A59-112 | Frank E Rheindt |
| Wildlife Reserves Singapore Conservation Fund | R-154-000-A99-592 | Frank E Rheindt |
| Croeni Foundation | R-154-000-A05-592 | Frank E Rheindt |
| SEABIG | R-154-000-648-646 | Balaji Chattopadhyay |
| SEABIG | R-154-000-648-733 | Balaji Chattopadhyay |
| University of Southampton | 511206105 | Kelvin S-H Peh |
| National Science Foundation | DEB-1441652 | Rauri CK Bowie |
| National Science Foundation | DEB-1457845 | Rauri CK Bowie |

The funders had no role in study design, data collection and interpretation, or the decision to submit the work for publication.

## Author contributions
Chyi Yin Gwee, Resources, Data curation, Formal analysis, Validation, Investigation, Visualization, Methodology, Writing - original draft, Project administration, Writing - review and editing; Kritika M Garg, Resources, Formal analysis, Investigation, Methodology, Project administration, Writing - review and editing; Balaji Chattopadhyay, Resources, Funding acquisition, Methodology, Project administration, Writing - review and editing; Keren R Sadanandan, Resources, Investigation, Project administration, Writing - review and editing; Dewi M Prawiradilaga, Resources, Project administration, Writing - review and editing; Martin Irestedt, Fumin Lei, Luke M Bloch, Mohammad Irham, Tri Haryoko, Malcolm CK Soh, Karen MC Rowe, Teuku Reza Ferasyi, Shaoyuan Wu, Resources, Writing - review and editing; Jessica GH Lee, Kelvin S-H Peh, Resources, Funding acquisition, Writing - review and editing; Guinevere OU Wogan, Resources, Formal analysis, Methodology, Writing - review and editing; Rauri CK Bowie, Resources, Funding acquisition, Investigation, Writing - review and editing; Frank E Rheindt, Conceptualization, Resources, Data curation, Formal analysis, Supervision, Funding acquisition, Validation, Investigation, Visualization, Methodology, Writing - original draft, Project administration, Writing - review and editing

## Author ORCIDs
Chyi Yin Gwee ⓘ https://orcid.org/0000-0003-1706-0520
Kelvin S-H Peh ⓘ http://orcid.org/0000-0002-2921-1341
Karen MC Rowe ⓘ http://orcid.org/0000-0002-6131-6418
Rauri CK Bowie ⓘ http://orcid.org/0000-0001-8328-6021
Frank E Rheindt ⓘ https://orcid.org/0000-0001-8946-7085

## Decision letter and Author response
Decision letter https://doi.org/10.7554/eLife.62765.sa1
Author response https://doi.org/10.7554/eLife.62765.sa2

# Additional files

## Supplementary files
• Supplementary file 1. Details of all samples included in the study. Abbreviations for museums: American Museum of Natural History, New York (AMNH); Australian National Wildlife Collection, Canberra (ANWC); Burke Museum of Natural History and Culture, Washington (Burke); Lee Kong Chian Natural History Museum, Singapore (LKCNHM); Museum of Vertebrate Zoology, California (MVZ); Museums Victoria, Melbourne, Australia (NMV); Museum Zoologicum Bogoriense, West Java, Indonesia (MZB); Naturalis Biodiversity Center, Leiden, Netherlands (Naturalis); South Australian Museum, Adelaide (SAMA); Swedish Museum of Natural History, Stockholm (NRM); Western Australian Museum, Perth (WAM); Yale Peabody Museum of Natural History, Connecticut (Peabody). Whole genome resequenced samples are marked with an asterisk (*) at the end of the sample ID.

• Supplementary file 2. A list of species assignable to one of the three main *Zosterops* clades (Indo-African, Asiatic, Australasian), either on the basis of our study (shaded in gray) or based on previous studies with a bootstrap support of at least 90. All species were assigned to their respective ranges of occurrence (see *Figure 1*): Afrotropical, West Indian Ocean (WIO), Asia, Philippines, Indonesian Archipelago (Indo), Melanesian Archipelago (Mel), Micronesian Archipelago (Mic), Australia and/or others.

• Transparent reporting form

## Data availability
All data generated or analysed during this study are included in Dryad database: https://doi.org/10.5061/dryad.8931zcrmt. Raw FASTQ files of target enriched samples are available on NCBI under BioProject no. PRJNA682287.

The following datasets were generated:

| | **Database and** |
| --- | --- |

| Author(s) | Year | Dataset title | Dataset URL | Identifier |
|---|---|---|---|---|
| Gwee CY, Garg KM, Chattopadhyay B, Sadanandan KR, Prawiradilaga DM, Irestedt M, Lei F, Bloch LM, Lee JGH, Irham M, Haryoko T, Soh MCK, Peh KSH, Rowe KMC, Ferasyi TR, Wu S, Wogan GOU, Bowie RCK, Rheindt FE | 2020 | Phylogenomics of white-eyes, a 'great speciator', reveals Indonesian archipelago as the center of lineage diversity | https://doi.org/10.5061/dryad.8931zcrmt | Dryad Digital Repository, 10.5061/dryad.8931zcrmt |
| Gwee CY, Garg KM, Chattopadhyay B, Sadanandan KR, Prawiradilaga DM, Irestedt M, Lei F, Bloch LM, Lee JG, Irham M, Haryoko T, Soh MC, Peh KSH, Rowe KM, Ferasyi TR, Wu S, Wogan GO, Bowie RC, Rheindt FE | 2020 | Phylogenomics of White-eyes Reveals Indonesian Archipelago as the Center of Lineage Diversity | https://www.ncbi.nlm.nih.gov/bioproject/PRJNA682287 | NCBI BioProject, PRJNA682287 |

The following previously published datasets were used:

| Author(s) | Year | Dataset title | Dataset URL | Database and Identifier |
|---|---|---|---|---|
| Cornetti L, Valente LM, Dunning LT, Quan X, Black RA, Hébert O, Savolainen V | 2015 | Zosterops lateralis melanops | https://www.ncbi.nlm.nih.gov/genome/40104?genome_assembly_id=248127 | NCBI Genome Assembly, 248127 |
| Leroy T, Anselmetti Y, Tilak MK, Bérard S, Csukonyi L, Gabrielli M, Scornavacca C, Milá B, Thébaud C, Nabholz B | 2019 | A bird's white-eye view on neosex chromosome evolution | https://www.ncbi.nlm.nih.gov/bioproject?LinkName=genome_bioproject&from_uid=17235 | NCBI BioProject, PRJNA530916 |

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
