## [Decision Letter]

**Acceptance summary:**

This study makes a valuable contribution to our understanding of the evolution of white-eyes, and more broadly the processes by which animals diverge and speciate. In particular, it highlights ways in which the biogeography of the Indonesian archipelago has generated the remarkable biodiversity of that region.

**Decision letter after peer review:**

Thank you for submitting your article "Phylogenomics of a 'Great Speciator' Reveals Indonesian Archipelago as the Center of Lineage Diversity" for consideration by *eLife*. Your article has been reviewed by three peer reviewers, and the evaluation has been overseen by a Reviewing Editor and George Perry as the Senior Editor. The following individual involved in review of your submission has agreed to reveal their identity: Leo Joseph (Reviewer #2)

We would like to draw your attention to changes in our revision policy that we have made in response to COVID-19 (https://elifesciences.org/articles/57162). Specifically, we are asking editors to accept without delay manuscripts, like yours, that they judge can stand as *eLife* papers without additional data, even if they feel that they would make the manuscript stronger. Thus the revisions requested below largely address clarity and presentation. In a few places, we suggest better justifying the current analyses and conclusions (e.g. explaining why an alternative analyses was not used).

All reviewers agreed that this study provides valuable insight into the phylogeography of *Zosterops*. It address the phylogenetic conflicts within the genus to resolve three main biogeographic clades. It further uses genomic data to provide evidence for introgression between species, highlighting that introgression is pervasive in rapidly diverging radiations. An impressive sample collection is used, spanning field and museum sampling.

Reviewers identified one main concern that should be addressed in a revision:

1) Borneo is posited as the origin of the radiation, but this notion could be better supported. Two suggestions were made:

i) Consider the region's tectonic history and include could some kind of modern biogeographic analysis, or a defense of why it is not included. For example, what about a BioGeoBears analysis?

ii) Consider alternative scenarios that might explain Borneo's species diversity; the apparent diversity in habitats of Borneo could have facilitated continental invaders to settle in the island. Perhaps, more clarification of the distribution of each species could provide better support to the authors' claims.

---

## [Author Response]

Reviewers identified one main concern that should be addressed in a revision:1) Borneo is posited as the origin of the radiation, but this notion could be better supported. Two suggestions were made:i) Consider the region's tectonic history and include could some kind of modern biogeographic analysis, or a defense of why it is not included. For example, what about a BioGeoBears analysis?

We have accommodated this concern in multiple ways: (1) We have added a BioGeoBears analysis that demonstrates an Asian origin for white-eyes and an Indonesian origin for specific white-eye clades. (2) We have added more discussion on Earth history, specifically the Quaternary history of Borneo with its changing land connections during the sea level fluctuations of the Pleistocene. (Please note that ‘tectonic’ history per se is less relevant to *Zosterops* as significant tectonic changes have occurred at a pace that’s much slower than the relatively recent diversification of this genus). (3) We have strengthened explanations throughout the manuscript to make clear that we are not focusing on Borneo as the origin of the radiation, but as the center of modern-day diversity of this genus.

ii) Consider alternative scenarios that might explain Borneo's species diversity; the apparent diversity in habitats of Borneo could have facilitated continental invaders to settle in the island. Perhaps, more clarification of the distribution of each species could provide better support to the authors' claims.

We have accommodated this concern in multiple ways: 1) We have added a BioGeo Bears analysis that demonstrates an Asian origin for white-eyes and an Indonesian origin for specific white-eye clades. 2) We have added more discussion on Earth history, specifically the Quaternary history of Borneo with its changing land connections during the sea level fluctuations of the Pleistocene. (Please note that ‘tectonic’ history per se is less relevant to *Zosterops* as significant tectonic changes have occurred at a pace that’s much slower than the relatively recent diversification of this genus). 3) We have strengthened explanations throughout the manuscript to make clear that we are not focusing on Borneo as the origin of the radiation, but as the center of modern-day diversity of this genus.